# Off-Target Effect of Activation of NF-κB by HIV Latency Reversal Agents on Transposable Elements Expression

**DOI:** 10.3390/v14071571

**Published:** 2022-07-20

**Authors:** Gislaine Curty, Luis P. Iniguez, Marcelo A. Soares, Douglas F. Nixon, Miguel de Mulder Rougvie

**Affiliations:** 1Oncovirology Program, Instituto Nacional de Cancer, Rio de Janeiro 20231-050, Brazil; gcf.science@gmail.com (G.C.); masoares@inca.gov.br (M.A.S.); 2Division of Infectious Diseases, Weill Cornell Medicine, New York, NY 10065, USA; lpr4001@med.cornell.edu (L.P.I.); dnixon@med.cornell.edu (D.F.N.); 3Department of Genetic Medicine, Weill Cornell Medicine, New York, NY 10065, USA

**Keywords:** transposable elements, human endogenous retroviruses, HERV, LINE-1, HIV-1, latency reversal agents, CD4+ T cells, T-cell subsets, PKC activation

## Abstract

Many drugs have been evaluated to reactivate HIV-1 from cellular reservoirs, but the off-target effects of these latency reversal agents (LRA) remain poorly defined. Transposable elements (TEs) are reactivated during HIV-1 infection, but studies of potential off-target drug effects on TE expression have been limited. We analyzed the differential expression of TEs induced by canonical and non-canonical NF-κB signaling. We evaluated the effect of PKC agonists (Bryostatin and Ingenol B) on the expression of TEs in memory CD4+ T cells. Ingenol B induced 38 differentially expressed TEs (17 HERV (45%) and 21 L1 (55%)). Interestingly, TE expression in effector memory CD4+ T cells was more affected by Bryostatin compared to other memory T-cell subsets, with 121 (107 upregulated and 14 downregulated) differentially expressed (DE) TEs. Of these, 31% (*n* = 37) were HERVs, and 69% (*n* = 84) were LINE-1 (L1). AZD5582 induced 753 DE TEs (406 HERV (54%) and 347 L1 (46%)). Together, our findings show that canonical and non-canonical NF-κB signaling activation leads to retroelement expressions as an off-target effect. Furthermore, our data highlights the importance of exploring the interaction between LRAs and the expression of retroelements in the context of HIV-1 eradication strategies.

## 1. Introduction

According to data from UNAIDS, over 37 million people were living with HIV-1 (PLWH) worldwide in 2020 [1]. Antiretroviral therapy (ART) is the principal approach implemented to control and suppress HIV-1 to undetectable levels to prevent AIDS [2,3]. However, HIV-1 is able to infect and develop a latent infection in CD4+ T cells, establishing a stable viral reservoir in memory CD4+ T cells. This is the main challenge for current HIV-1 cure strategies [4,5,6]. Therapeutic strategies are focused on how to eliminate the HIV-1 reservoir, one approach is known as the “shock and kill strategy”, in which latency reversal agents (LRA) reactivate HIV-1 from cellular reservoirs (shock) for elimination by ART treatment (kill) [7,8]. However, LRAs are not specific for HIV-1 reactivation, and they can also impact on the expression of other genes such as transposable elements (TE) [9,10,11]. 

Over 40% of the human genome is composed of ancient TE. They are divided into retroelements and transposons. Retroelements are endogenous sequences which replicate through an RNA intermediate, and are divided into those with long terminal repeat (LTR) or without (non-LTR) [12]. LTR retrotransposons are old virus sequences integrated in the genome and they compose about 8% of the human genome and are known as human endogenous retroviruses (HERV) [12,13]. Non-LTR retrotransposons represent about 33% of the genome and are divided into short interspersed nuclear elements (SINE) and long counterparts (LINE or L1) [12]. While SINEs are elements which depend on L1 expression for their replication, L1s are autonomous elements able to replicate into the human genome. In addition, HERV and L1 expression in cells are associated with viral infections and human diseases, such as HIV-1 and cancer [14,15,16,17,18,19]. 

Differential HERV and L1 expression has been reported during LRA treatment [9,10]. However, the effect of newer LRAs such as bryostatin-1 and ingenol-3-hexanoate (Ingenol B) on TE expression has not been investigated. Both induce gene expression though PKC-NF-κB signaling and are able to induce HIV expression [20,21,22]. They promote phosphorylation and inactivation of IκB proteins, allowing for NF-κB translocation to the nucleus and the initiation of gene transcription. In addition, phorbol 12-myristate 13-acetate (PMA), another agonist of PKC used to activate T cells, can also stimulate HIV-1 expression through PKC–NF-κB signaling [23,24]. In addition, a new class of LRAs known as mimetics of the second mitochondrial-derived activator of caspases (SMACm—SMAC mimetics) are also able to induce HIV-1 expression through non-canonical NF-κB signaling [21,25]. They are able to induce HIV latency reversion alone or combined with other LRAs [26]. SMAC mimetics antagonize the inhibitor of apoptosis proteins (IAPs) such as AZD5582, which binds to the BIR3 domains cIAP1, cIAP2 and XIAP and induces apoptosis [27]. The cIAP1/cIAP2 degradation induces non-canonical NF-κB signaling activation in resting CD4+ T cells from PLWH on ART and in latently infected cells from models of HIV-1 latency. In addition, the activation of non-canonical NF-κB signaling by AZD5582 induces HIV-1 RNA expression in the blood of ART-suppressed bone marrow–liver–thymus in humanized mice [25]. Although several studies have explored the effects of LRAs on HIV-1 reactivation, little is known about their impact on TE expression. In this study, we evaluated the off-target effects of two PKC agonists (Bryostatin-1 and Ingenol B) and a SMAC mimetic (AZD5582) on the expression of TEs in memory CD4+ T-cell subsets.

## 2. Materials and Methods

### 2.1. RNA-seq Datasets

In this study, we used publicly available RNA-seq datasets to analyze the impact of TE expression in T-cell subsets treated with two different PKC LRAs. Datasets were downloaded from the Sequence Read Archive (SRA) with the following accession numbers: GSE142774 and GSE94150. We selected studies where LRAs (Bryostatin, Ingenol-B or AZD5582) were used ex vivo on memory CD4+ T cells obtained from PLWH. In GSE142774, resting CD4+ T cells obtained from HIV-1-infected and ART-suppressed donors were treated ex vivo with PKC agonists (Ingenol B) and SMACm (AZD5582). The activity of Ingenol B (25 nM), AZD5582 (100 nM) or DMSO (0.05%) on memory CD4+ T cells was evaluated, and cells were harvested at 2, 6, and 24 h after exposure. RNA was isolated from the harvested cells, processed into stranded mRNA libraries and then sequenced using an Illumina Hiseq 4000 sequencer using a paired-end 50 bp × 50 bp run [25]. In GSE94150, CD4+ T cells were isolated from PLWH who had been on successful ART for over 36 months with an undetectable viral load (≤50 copies/mL). T cells were sorted into three T-cell subpopulations: central memory (CM), effector memory (EM) and transition memory (TM) cells. The ex vivo effects of PKC agonists (Bryostatin and PMA) were assessed for their ability to induce viral transcription in sorted memory CD4+ T-cell subsets following a 24 h stimulation with 10 nM Bryostatin and 100 ng/mL PMA. Total RNA was isolated, converted into cDNA, and libraries were generated using the Nextera XT DNA library preparation kit and sequenced on an Illumina HiSeq 2500 [28]. In short, these are ex vivo studies using memory CD4+ T-cell subsets obtained from ART-suppressed HIV-infected individuals. We used RNA-seq data obtained from CD4+ T-cell subsets treated for the same 24 h period with an array of LRAs: Bryostatin, Ingenol B and AZD5582. DMSO and PMA were used for drug control comparison. Following LRA treatment, RNA was isolated, and libraries were prepared and sequenced on the Illumina platform.

### 2.2. Bioinformatic Analyses

FASTQ files from RNA datasets were downloaded in the GEO and aligned to the human genome (hg38) using Bowtie2 (very-sensitive-local-k 100—score-min L, 0, 1.6 for multi-mappings) [29,30]. The Bowtie2 output was used in the Telescope software. Telescope was used to define and quantify accurate retrotransposon expression on samples using HERV and L1 annotation (retro.hg38.v1, available on https://github.com/mlbendall/telescope_annotation_db/tree/master/builds, accessed on 19 July 2022) [31]. Host gene quantification was performed using Salmon v1.3.0 through gencode v.33 plus the HIV-1 genome (K03455) [32]. Telescope and Salmon outputs were used to calculate retrotransposons differentially expressed in treated vs. untreated CD4+ T cells using DESeq2 using the Wald test [33]. Retrotransposons with adjusted *p*-values < 0.05 and absolute (log2FoldChange) > 1.5 were considered differentially expressed and visualized using Volcano plots with Bioconductor EnhancedVolcano (https://github.com/kevinblighe/EnhancedVolcano, accessed on 19 July 2022) and by the R ggplot2 package. Unsupervised hierarchical analysis using retroelements abundance data was performed using Bray–Curtis dissimilarity and average linkage metrics by the R vegan package. In addition, Venn and upset plots were performed using the Venn diagram and UpSetR packages in R to show all relationships between sets of data from differentially expressed retroelements. The most expressed HERV and L1 were shown by bar plots. HERV families were defined according to the retro.hg38.v1 annotation and shown by pie charts. Neighboring host genes from differentially expressed retroelements and whether the TEs were noncoding or coding were defined using Telescope meta annotations (https://raw.githubusercontent.com/LIniguez/Telescope_MetaAnnotations/main/TE_annotation.v2.0.tsv, accessed on 19 July 2022). Over-representation analysis (ORA) was also performed with WebGestal (available at http://webgestalt.org/, accessed on 19 July 2022) using gene ontology biological processes as a functional database.

## 3. Results

In order to characterize the profile of TE expression in CD4+ T cells upon distinct LRA treatments, we first evaluated the off-target ex vivo effect of canonical NF-κB activation induced by PKC signaling from Ingenol B and Bryostatin in CD4+ T cells obtained from aviremic HIV-1 participants. We found a total of 38 differentially expressed (23 upregulated and 15 downregulated) transposable elements in memory CD4+ T cells treated with Ingenol B (Figure 1A). Of these, 21 (55.3%) were assigned to L1 sequences and 17 (44.7%) to HERV sequences. Among the most differentially expressed HERV loci we found: MER41_14q24.2a, ERV316A3_2q14.1d, PRIMA41_19q13.2b, PRIMA41_19q13.2a, HERV4_4q12, HERVP71A_3p26.1, HERV30_6q22.31, ERVLB4_11p15.5d, HERVK11D_2q11.2 and HERVH_11q24.1a (Figure 1B). Regarding DE L1 loci, we detected: L1FLnI_12p12.1q, L1FLnI_1q32.2m, L1FLnI_12p12.1r, L1FLnI_12p12.1s, L1FLnI_Xq22.1la, L1FLnI_4q25l, L1FLnI_13q22.1c, L1FLnI_5p15.33b, L1FLnI_5q35.3b and L1FLnI_13q22.1d (Figure 1C).

We also evaluated the effect of PKC activation induced by Bryostatin on the expression of TEs in central memory (CM), transition memory (TM) and effector memory (EM) CD4+ T-cell subsets. Interestingly, EMs were the most impacted cell population by Bryostatin compared to other CM and TM subsets. We found a total of 121 (14 upregulated and 107 downregulated) differentially expressed TEs in the Bryostatin treatment (Figure 2A). Of these, 30.6% (*n* = 37) were HERVs and 69.4% (*n* = 84) were L1 sequences. The most upregulated TEs induced by Bryostatin treatment were: L1FLnI_2p25.3c, HARLEQUIN_8p11.21, ERVLB4_2q37.1a, HUERSP3_1p36.13, L1FLnI_10q23.33a. The most downregulated TEs were L1FLnI_12p13.31k, L1FLnI_7q34b, L1FLnI_Xp11.21ec, L1FLnI_11p11.2c and L1FLnI_17p13.1a. The number of differentially expressed TEs following Bryostatin treatment was 113 and 4 in CM and TM cell populations, respectively (Figure 2A). Of the 113 differentially expressed TEs in CM, we found 49 up- and 64 downregulated in Bryostatin-treated cells. Of these, 35.4% (*n* = 40) were HERVs and 64.6% (*n* = 73) were L1 sequences. Regarding differentially expressed TEs in TM cells, we only detected upregulated L1 sequences induced by Bryostatin treatment (L1FLnI_5q21.3p, L1FLnI_19q13.12c, L1FLnI_13q34c and L1FLnI_5q23.1c). Additionally, we also evaluated the effect of phorbol myristate acetate (PMA) on T-cell subsets (Figure 2B). PMA is known to promote elevated levels of T cells’ activation through the PKC activation signal pathway and HIV expression. In EM T cells, a total of 150 differentially expressed TEs (112 up- and 38 downregulated) were detected following PMA treatment (Figure 2B). Of those, 67.3% (*n* = 101) were L1 sequences and 32.7% (*n* = 49) were HERV sequences. In CM T cells, we found 89 differentially expressed TEs (63 up- and 26 downregulated sequences) (Figure 2B). Of these, 35 were HERVs and 54 were L1 sequences. In TM T cells we found 85 differentially expressed TEs upon PMA treatment (70 up- and 15 downregulated to PMA treatment), of which 27 (31.8%) were HERVs and 58 (68.2%) were L1 sequences (Figure 2B).

Most of the differentially expressed TEs were exclusively regulated for each memory T-cell subset tested; 103, 95 and 3 TEs were exclusively modulated in EM, CM and TM cells treated with Bryostatin, respectively (Figure 2C). Most interestingly, 17 DE TEs (5 HERVs and 12 L1) such as ERV316A3_21q22.11b, ERVLE_17q25.3a, HUERSP1_Xp22.2, MER101_18q21.1b and MER41_21q22.11b were common to CM and EM; and one L1, localized on chromosome 13 (L1FLnI_13q34c), was common to all T-cell subsets. In addition, 15, 7 and 11 differentially expressed TEs were found commonly regulated between CM and EM, CM and TM, and EM and TM, respectively (Figure 2D). In addition, two DE TEs (HUERSP2_19q13.2 and L1FLnI_Xq23d) were common to all T-cell subsets. Among all DE TEs, 65, 65 and 122 of them were exclusively regulated in TM, CM and EM cells treated with PMA, respectively.

Next, we compared the effect on TE expression caused by different LRA PKC agonists (Bryostatin and Ingenol B) in treated memory CD4+ T-cell subsets. We found L1FLnI_Xq22.1la to be commonly upregulated in EM and CM T cells treated with Bryostatin and in general memory T cells treated with Ingenol B (Figure 2E). Furthermore, some differentially expressed TEs were exclusively regulated by LRA treatment according to each memory CD4+ T-cell subset. A total of 39, 8 and 2 TEs were found exclusively upregulated in CM, EM and TM cells treated with Bryostatin, respectively. In memory CD4+ T cells we found 20 TEs exclusively expressed by Ingenol B treatment. In addition, HERV loci from full-length HERV families were found exclusively up-regulated in memory CD4+ T cells treated with Ingenol B HERV9_1p36.23(HERVW), HML3_1p36.23 (HERVK) and in CM cells treated with Bryostatin HERV-K (HML2_14q11.2, HML3_5p15.33b, HML3_Xq13.3) (Figure 2E). Together, these data show that PKC agonists lead to differential TE expression in each memory CD4+ T cell.

We also analyzed TE expression in noncanonical NF-κB activation from memory CD4+ T cells treated with AZD5582 and compared those with Ingenol B and Bryostatin treatment, which are also able to induce canonical NF-kB activation. In an unsupervised analysis we found three clusters from TE loci expressed in cells treated with DMSO, Ingenol B and AZD5582 (Figure 3A). We found 753 differentially expressed TEs (406 HERV and 347 L1) in AZD5582 (Figure 3B). Two full-length HERVs, such as HERVL (9p24.1b and 5q14.3e loci) and L1 sequences localized on chromosome 1 (L1FLnI_1p13.1f), 9 (L1FLnI_9q13g), 5 (L1FLnI_5q31.1k), 6 (L1FLnI_6p23) and 19 (L1FLnI_19q13.42b) were upregulated by AZD5582 treatment (Figure 3C). In contrast, HERVK (HML2_1q32.2) and HERVH (HERVH_8q24.21a) were the most downregulated loci in cells treated with AZD5582 (Figure 3C). Comparison between the HERV families regulated between the AZD5582 and Ingenol B treatment shows that the Harlequin, HERVE, HERVI, HERVL, HERVS, HUERS, LTR and PAB families were exclusively regulated in cells treated with AZD5582 (Figure 3D). On the other hand, the ERV1 and HERVP families were differently regulated between the AZD5582 and Bryostatin treatment (Figure 3D).

In addition, we found three L1 loci, localized in the chromosomes 16, 17 and X, commonly upregulated by treatments (Figure 3E). Additionally, 140 TEs (66 HERV and 74 L1 loci), 13 TEs (7 HERV and 6 L1 loci) and 44 TEs (15 HERV and 29 L1 loci) were upregulated exclusively by AZD5582, Ingenol B and Bryostatin treatment. The retroelements’ neighboring genes and the either coding and noncoding DE TE data were also defined to each type of treatment (Appendix A). There were no significant enriched pathways of the neighboring genes from DE TE (Appendix A). Altogether, these data show that LRAs can regulate the differential expression of retroelements through canonical and non-canonical NF-κB activation (Figure 4).

## 4. Discussion

The HIV genome can be stabilized into cellular reservoirs through latent infection and can persist for decades in the human body. Thus, many drugs have been evaluated to reactivate and eliminate HIV-1 from reservoirs but their transcriptional reactivation is non-specific, and off-target effects have been reported [9,10,11,34,35]. In this study, we analyzed the off-target effects on the expression of retroelements in memory T cells treated with Bryostatin, Ingenol B and AZD5582, which are able to induce NF-κB signaling pathways. 

Bryostatin-1 and Ingenol B lead to HIV-1 reactivation in vivo and ex vivo. Bryostatin-1 has also been assessed in clinical trials to treat Alzheimer and cancer [36,37]. Both drugs activate the PKC-NF-κB signaling pathway and induce gene expression. The phosphorylation and inactivation of IκB proteins leads to the translocation of NF-κB to the nucleus and induces HIV-1 and endogenous gene transcription [21]. Herein, we found a total of 38 differentially expressed transposable elements in memory CD4+ T cells treated with Ingenol B. The MER4 (14q24.2a loci), ERV3 (2q14.1d loci) and PRIMA (19q13.2b and 19q13.2a loci) families were the most upregulated HERVs by Ingenol B in T cells. Additionally, L1 sequences from chromosome 1 (1q32.2m), 12 (12p12.1q, 12p12.1r and 12p12.1s) and X (Xq22.1la) were also found to be the most upregulated in treated cells. The expression of transposable elements might induce immune responses, genomic instability, tumorigenesis and can be used as biomarkers, which could suggest a functional effect role in memory T cells treated with LRA [15,38,39]. Memory CD4+ T cells are classified into central, transitional and effector memory cells, and each of them have a distinct cell phenotype, function and survival properties [40,41,42]. Although all memory CD4+ T-cell subsets have been shown to contribute to HIV-1 reservoirs, EM cells have the largest proportion of HIV-1 intact genomes and show the highest latency reversal phenotype compared to either CM or TM subsets [28,43]. Here, we found differential TE regulation in memory CD4+ T-cell subsets (CM, TM and EM) through PKC activation mediated by Bryostatin. EM cells were the most impacted cell population compared to other memory CD4+ T-cell subsets. A total of 103, 95 and 3 TEs were exclusively modulated in EM, CM and TM cells treated with Bryostatin, respectively. In addition, classical latency reversal attempts are achieved through T-cell activation, thus we also evaluated the effect of TE expression in memory T-cell subsets treated with PMA, a positive control of T-cell activation mediated by PKC activation [23,24]. Similar to Bryostatin, the EM subpopulation was the most affected cell subset, showing a total of 150 differentially expressed TEs (112 up- and 38 downregulated) following PMA treatment. Together, our data show a distinct TE expression pattern in each memory CD4+ T-cell subset, suggesting a differential molecular regulation from the EM-cell subset treated with LRA. Similarly, a study analyzing the transcriptional profiling of host genes from CM, TM and EM cells has shown a distinct response to Bryostatin, which might be associated with signaling pathways specific to each subset [28]. That same study has also shown the expression of several genes associated with greater LRA responsiveness in EM cells, such as PKC activation pathway genes, NF-κB and the nuclear factor of activated T-cell (NFAT) pathway genes as well as AP-1 and E2F transcription factor target and proinflammatory pathway genes [28].

We also compared the effect of two LRA PKC agonists (Bryostatin and Ingenol B) on the regulation of TE expression in memory T cells. We found L1 sequences located in chromosome X (L1FLnI_Xq22.1la) to be commonly upregulated in EM and CM T cells treated with Bryostatin and, in general, in memory T cells treated with Ingenol B. Prior studies have shown how L1 sequences integrated into the X chromosome are different to those integrated in autosomes. The X chromosome is enriched with young L1 sequences (inserted <100 million years ago) which are involved in X inactivation [44,45,46,47,48]. We also found a total of 39, 8 and 2 TEs exclusively upregulated in CM, EM and TM cells treated with Bryostatin, respectively. In addition, 20 TEs were exclusively expressed in memory T cells treated with Ingenol B. Furthermore, full-length HERV loci were found exclusively upregulated in T cells treated with Bryostatin (HERV-K (HML2_14q11.2, HML3_5p15.33b and HML3_Xq13.3 loci) and with Ingenol B (HERVW (HERV9_1p36.23 locus) and HERVK (HML3_1p36.23 locus). Interestingly, full-length HERV elements such as HERV-K and HERV-W display an integrated complete genome (~9.5 kb) with two long terminal repeats (LTRs) that flank three structural viral genes (*gag*, *pol* and *env*) encoding Env, PR, RT and Gag proteins, which have been shown to be able to modulate cellular pathways, stimulate immune responses and reduce the infectivity of HIV-1 [19,39,49,50,51,52]. These findings show that PKC activator drugs lead to differential TE expression in each memory CD4+ T-cell subset, which can lead to a distinct cellular phenotype depending on the LRA treatment.

Moreover, we also analyzed differential TE expression in noncanonical NF-kB activation induced by AZD5582. SMACm, such as AZD5582, is a new class of LRAs that leads to cIAP1/cIAP2 degradation and induces non-canonical NF-κB signaling activation and apoptosis. AZD5582 potently activates the NFκB pathway, but does not induce apoptosis in non-tumor cells and shows a modest off-target effect in host gene expression in CD4+ T cells treated ex vivo when compared with a stabilized Ingenol-B derivative (GSK445A) [53,54,55]. Herein, we found 753 differentially expressed TEs (406 HERV and 347 L1) in AZD5582-treated cells. In addition, comparison between the HERV families shows that Harlequin, HERVE, HERVI, HERVL, HERVS, HUERS, LTR, PAB, ERV1 and HERVP were exclusively regulated in AZD5582-treated cells. These results suggest a putative TE-related molecular marker that could be used to detect HIV-1 infected cells following NF-κB activation. Finally, taken together, we show that LRAs can regulate the differential expression of retroelements through canonical and non-canonical NF-κB activation, suggesting that they can lead to distinct cellular phenotypes, which might be used as biomarkers and/or immunotherapeutic targets to eliminate specific memory CD4+ T-cell subsets involved in HIV-1 persistence.

## Figures and Tables

**Figure 1 viruses-14-01571-f001:**
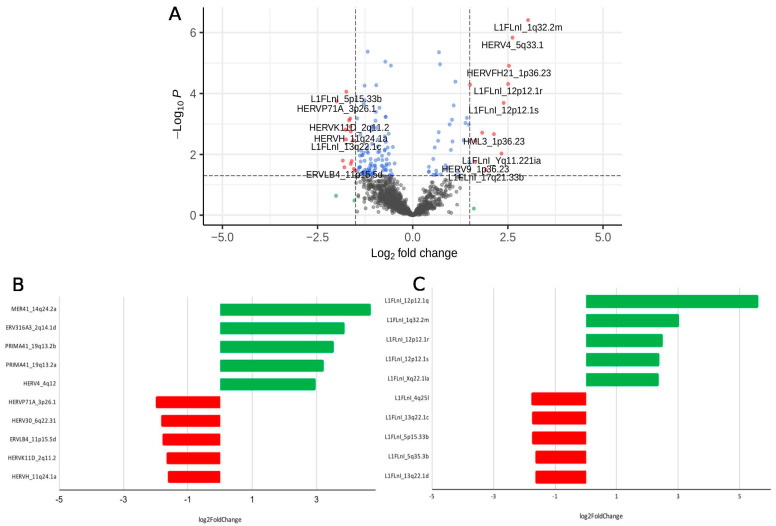
Differential expression of transposable elements in CD4 T cells treated with Ingenol B. (**A**) Volcano Plot of the comparison Ingenol B vs. DMSO; Top differentially expressed HERV (**B**,**C**) LINE-1 loci.

**Figure 2 viruses-14-01571-f002:**
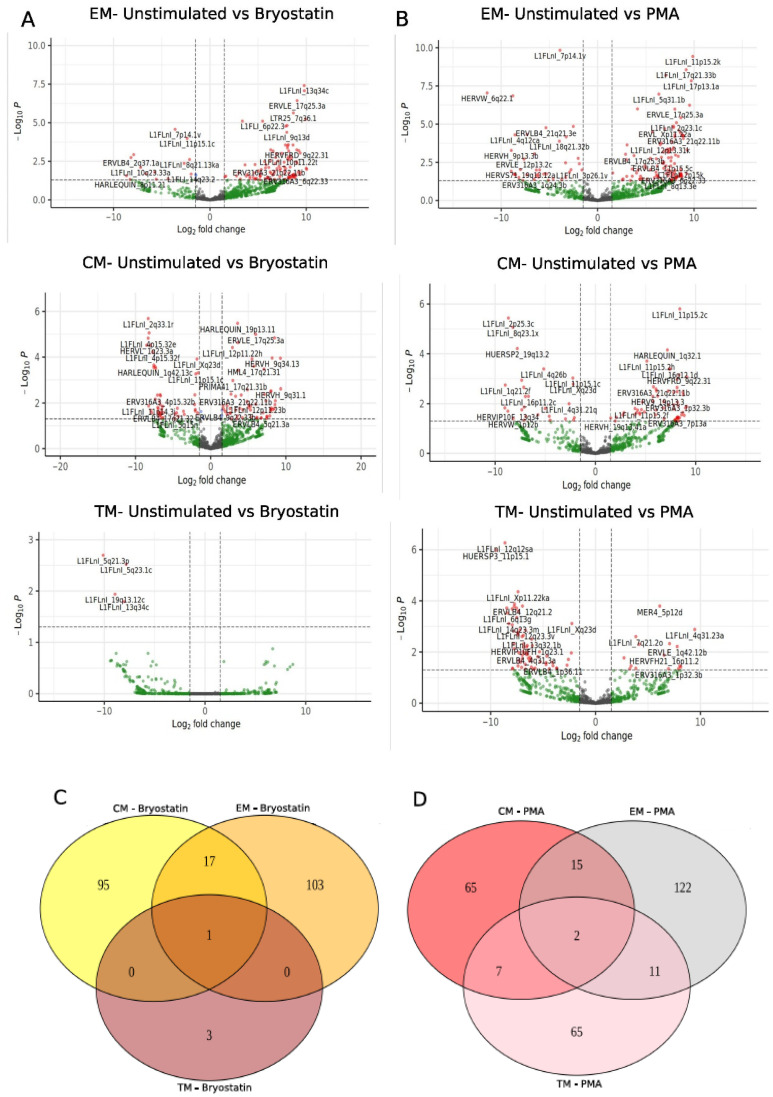
Differential expression of transposable elements in CD4+ T cells treated with Bryostatin and PMA. Volcano plots for each memory CD4+ T-cell subset are shown for Bryostatin (**A**) and PMA treatment (**B**). Venn plots depict commonly and exclusively regulated TEs for Bryostatin (**C**) and PMA (**D**). (**E**) UpSet plot for up- and downregulated TEs in memory T CD4+ treated with Bryostatin and Ingenol B.

**Figure 3 viruses-14-01571-f003:**
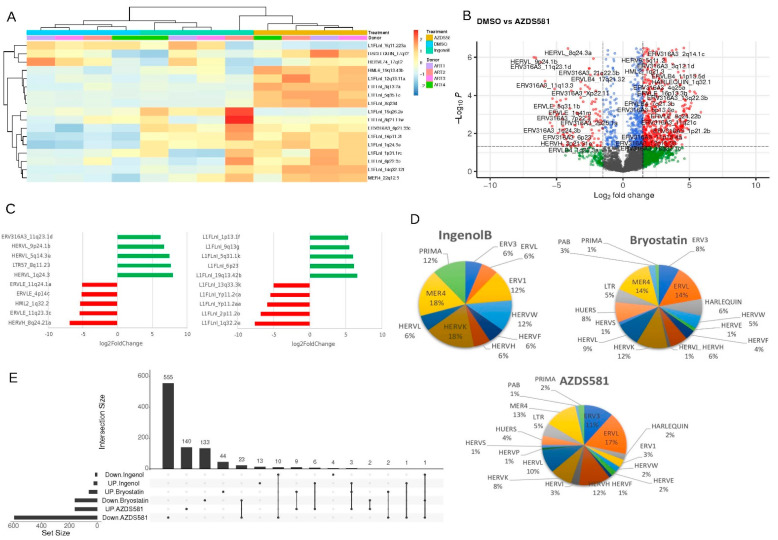
Canonical and noncanonical NF-κB activation from memory CD4+ T cells. (**A**) Unsupervised heatmap of most expressed TE by each LRA treatment (Ingenol B or AZDS581) with relative abundance ≥1%. Labels from donors and kind of treatment are also plotted. (**B**) Volcano plot of differential TE expressed by DMSO vs. AZDS581 treatment. Red dots are significantly regulated TE in the AZDS581 treated cells. (**C**) Bar plot of the most differential HERV and L1 expressed showed upregulated (green bar) and downregulated (red bar) TEs by AZDS581 treatment. (**D**) Pie chart shows HERV families regulated in AZD5582, Ingenol B and Bryostatin treatment. (**E**) UpSet plot shows up- and downregulated TEs by each treatment.

**Figure 4 viruses-14-01571-f004:**
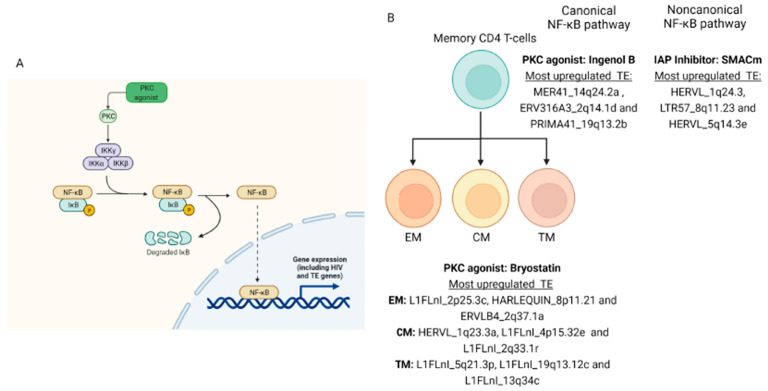
TE expression by PKC LRA. (**A**) Canonical NF-κB pathways lead to TE expression in the PKC activation during HIV shock and kill strategy and (**B**) most differential TE expressed in memory CD4+ T cells.

## Data Availability

Datasets were downloaded from the Sequence Read Archive (SRA) with the following accession numbers: GSE142774 and GSE94150.

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
