# Peer review of "Off-Target Effect of Activation of NF-κB by HIV Latency Reversal Agents on Transposable Elements Expression"

_viruses, 2022, doi:10.3390/v14071571_

Round 1
Reviewer 1 Report
Transposable elements (TEs) are reactivated during HIV-1 infection. Many drugs have been evaluated to reactivate HIV-1 from cellular reservoirs. There is few studies of potential off-target drug effects on TE expression. In this paper, the authors show that canonical and non-canonical NF-κB signaling activation leads to retroelement expressions as an off-target effect.
This is an interesting study.
1. In the abstract, the authors showed Ingenol B induced 39 differentially expressed TEs(17 HERV (45%) and 21 L1 (55%)). 17 plus 21 is not 39, it is 38. The authors should explain this.
2. The authors analyzed TE expression in noncanonical NF-kB activation from memory 195 CD4+ T cells treated with AZD5582 and compared those with Ingenol B. I suggested the authors should also compare AZD5582 with Bryostati.
Author Response
Reviewer 1
We thank the reviewer for his/her input and appreciate the time taken to improve the quality of this manuscript.
Transposable elements (TEs) are reactivated during HIV-1 infection. Many drugs have been evaluated to reactivate HIV-1 from cellular reservoirs. There is few studies of potential off-target drug effects on TE expression. In this paper, the authors show that canonical and non-canonical NF-κB signaling activation leads to retroelement expressions as an off-target effect.
This is an interesting study.
- In the abstract, the authors showed Ingenol B induced 39 differentially expressed Tes (17 HERV (45%) and 21 L1 (55%)). 17 plus 21 is not 39, it is 38. The authors should explain this.
Thank you for noticing this error. We have corrected the abstract to 38 DE TE (17 HERV + 21 L1).
- The authors analyzed TE expression in noncanonical NF-kB activation from memory CD4+ T cells treated with AZD5582 and compared those with Ingenol B. I suggested the authors should also compare AZD5582 with Bryostatin.
Thank you for this suggestion. In the revised version of the manuscript, we added this new comparison and described it in the results section. We have also updated figure 3 to include these results.
________________________________________________________________________
Reviewer 2 Report
The article “Off-target of activation of NF-kB by HIV latency reversal agents on transposable elements expression” by Gislaine Curty, Mulder Rougvie, and coworkers is an interesting research article. One important message is that LRA for HIV-1 therapy activates the transposable elements. Interestingly, the expression pattern of retroelements is different among the memory CD4+T cells. The differential TE expression might be biomarkers and/or immunotherapeutic targets to kill the latent HIV-1-infected T cells.
I have some questions about the manuscript.
In the experiment shown in Figure 2, TE expression pattern is the difference among the memory CD4+T cells. However, I am concerned about whether the differential expression of TEs in CD4+T cells is reproducible. Is it possible that the activation of transposable elements might be random?
The most differential TEs is presented in Figure 4B. Can you describe the feature of these TEs in detail? For example, the open reading frame of viral proteins, neighboring genes, encoded in intragene? or intergene? and so on.
Minor
On page 2, line51, there is another article about the association between HERV and HIV infection. The activated-HERV expression by LRA might interfere with the HIV-1 replication. If the full-length of HERV-K is activated by Bryostatin and Ingenol B as described in the discussion part (lines 279-286), please cite the article (Monde, J.Virol 2012).
Page 4, line 143, please modify “69,4%” to “69.4%”.
Author Response
Reviewer 2
We appreciate the reviewer’s time and efforts to improve the submitted manuscript.
The article “Off-target of activation of NF-kB by HIV latency reversal agents on transposable elements expression” by Gislaine Curty, Mulder Rougvie, and coworkers is an interesting research article. One important message is that LRA for HIV-1 therapy activates the transposable elements. Interestingly, the expression pattern of retroelements is different among the memory CD4+T cells. The differential TE expression might be biomarkers and/or immunotherapeutic targets to kill the latent HIV-1-infected T cells.
I have some questions about the manuscript.
In the experiment shown in Figure 2, TE expression pattern is the difference among the memory CD4+T cells. However, I am concerned about whether the differential expression of TEs in CD4+T cells is reproducible. Is it possible that the activation of transposable elements might be random?
This is an important question. As we mention in the discussion section of the manuscript, the literature shows that memory CD4+ T-cells (central, transitional, and effector memory cells) display distinct cell phenotype, function, and survival properties. For example, EM cells have the largest proportion of HIV-1 intact genomes and show the highest latency reversal phenotype compared to either CM and TM subsets. Following this, previous findings agree with our data, EM cells were the most impacted cell population compared to other memory CD4+ T-cells subsets regarding retroelement regulation. Therefore, the distinct cell phenotype and function of these cells might be responsible for the difference in the retroelements expression observed in different memory T cells.
The most differential TEs is presented in Figure 4B. Can you describe the feature of these TEs in detail? For example, the open reading frame of viral proteins, neighboring genes, encoded in intragene? or intergene? and so on.
Thank you for your suggestion. In the modified version of the manuscript, we included Supplementary Figure 1 that displays information on enriched neighbor genes as well as coding and noncoding retroelements data.
Minor
On page 2, line51, there is another article about the association between HERV and HIV infection. The activated-HERV expression by LRA might interfere with the HIV-1 replication. If the full-length of HERV-K is activated by Bryostatin and Ingenol B as described in the discussion part (lines 279-286), please cite the article (Monde, J.Virol 2012).
Done.
Page 4, line 143, please modify “69,4%” to “69.4%”.
Done.
________________________________________________________________________
Round 2
Reviewer 1 Report
well revised.
Reviewer 2 Report
I think that the manuscript is well written.
Only one minor mistake.
Page 8, line 217. "The of retroelements...." Please check this sentence.